# HMGXB4 Targets *Sleeping Beauty* Transposition to Germinal Stem Cells

**DOI:** 10.3390/ijms24087283

**Published:** 2023-04-14

**Authors:** Anantharam Devaraj, Manvendra Singh, Suneel A Narayanavari, Guo Yong, Jiaxuan Chen, Jichang Wang, Mareike Becker, Oliver Walisko, Andrea Schorn, Zoltán Cseresznyés, Tamás Raskó, Kathrin Radscheit, Matthias Selbach, Zoltán Ivics, Zsuzsanna Izsvák

**Affiliations:** 1Max-Delbrück-Center for Molecular Medicine in the Helmholtz Society (MDC), Robert-Rössle-Strasse 10, 13125 Berlin, Germany; 2Division of Hematology, Gene and Cell Therapy, Paul-Ehrlich-Institute, Paul-Ehrlich-Strasse 51-59, 63225 Langen, Germany

**Keywords:** *Sleeping Beauty*, transposon, HMGXB4, transcriptional activator, H3K4me3, NuRF, MLL complex, chromatin remodeling, nucleolus, BAP18/C17orf49, SUMOylation, germline, germinal stem cell, Wnt signaling, chromatin domain boundary

## Abstract

Transposons are parasitic genetic elements that frequently hijack vital cellular processes of their host. HMGXB4 is a known Wnt signaling-regulating HMG-box protein, previously identified as a host-encoded factor of *Sleeping Beauty* (SB) transposition. Here, we show that HMGXB4 is predominantly maternally expressed, and marks both germinal progenitor and somatic stem cells. SB piggybacks HMGXB4 to activate transposase expression and target transposition to germinal stem cells, thereby potentiating heritable transposon insertions. The *HMGXB4* promoter is located within an active chromatin domain, offering multiple looping possibilities with neighboring genomic regions. *HMGXB4* is activated by ERK2/MAPK1, ELK1 transcription factors, coordinating pluripotency and self-renewal pathways, but suppressed by the KRAB-ZNF/TRIM28 epigenetic repression machinery, also known to regulate transposable elements. At the post-translational level, SUMOylation regulates HMGXB4, which modulates binding affinity to its protein interaction partners and controls its transcriptional activator function via nucleolar compartmentalization. When expressed, HMGXB4 can participate in nuclear-remodeling protein complexes and transactivate target gene expression in vertebrates. Our study highlights HMGXB4 as an evolutionarily conserved host-encoded factor that assists *Tc1/Mariner* transposons to target the germline, which was necessary for their fixation and may explain their abundance in vertebrate genomes.

## 1. Introduction

Transposons or transposable elements (TEs) are discrete DNA segments that can move and replicate within genomes across the tree of life. While TEs can invade new “naïve” genomes by horizontal transfer [1], since somatic transposition is not heritable, the transfer can only be successful if the TE gets fixed in the germline of the targeted genome. 

To achieve heritable mobilization, certain TEs transpose in undifferentiated germ cells (primordial germ cells) during embryonic and larval stages and germinal stem cells in later developmental stages. This strategy is used by the *P-element* in *Drosophila* and controlled by the Piwi-interacting small RNA (piRNA) pathway [2]. In contrast, retrotransposons barely mobilize directly in germline stem cells [3], and use an alternative strategy. In this scenario, certain *Drosophila* retrotransposons were shown to “hijack” the microtubule transporting system to transfer their transcripts from the interconnecting supporting nurse cells to the transcriptionally inactive oocyte [3]. Mammalian retrotransposons likely use a similar scheme [4]. Curiously, TEs might even manipulate the blastomere to switch between germinal and somatic fates. In conjunction with this, TE-derived sequences have been incorporated into gene regulatory networks of the pluripotent cells [5,6,7,8,9]. 

The horizontal transfer could be supported by host-encoded factors, assisting the transposition mechanism. TEs (and viruses) frequently piggyback host-encoded factors to assist their life cycle. These factors often play essential roles in the host organism and are usually phylogenetically conserved, thus are readily available in the naïve organism. This work is based on the assumption that HMGXB4 is such a candidate. 

Previously, we had identified HMGXB4 as a host-encoded factor for *Sleeping Beauty* (SB) transposition, serving as a transcriptional activator of transposase expression [10]. SB was resurrected from inactive transposon copies from various fish genomes [11]. SB can transpose in a wide variety of cells, including both somatic and germinal origin [12]. However, how SB is targeted to the germline is unknown. SB transposes via a DNA-based “cut and paste” mechanism and utilizes several conserved host-encoded factors [10,13,14,15,16,17]. These host-encoded factors regulate transposition throughout the transposition process [17]. In the SB transposon, the transposase gene is flanked by two terminal inverted repeats (IRs), which carry recognition motifs for the transposase. HMGXB4 enhances transposase expression by interacting with sequences located in the 5′-UTR region of the transposon that function as a promoter of the transposase [10]. Following the premise that HMGXB4 is involved in key biological processes, attractive to be captured by a transposon, our SB transposition model is also suitable to uncover these host-encoded functions. 

HMGXB4 (previously known as HMG2L1) was shown to inhibit Wnt signaling [18] and differentiation [19]. Nevertheless, HMGXB4 is not commonly recognized as relevant for development. HMGXB4 was also detected as the most abundant protein in the interactome of the ATP-dependent nucleosome remodeling NuRF (nucleosome remodeling factor) complex [20]. Still, its role in chromatin remodeling is not characterized. 

NuRF is a phylogenetically conserved chromatin remodeling complex initially identified in *Drosophila* [21]. The multi-subunit NuRF complex relaxes the condensed chromatin to promote DNA accessibility and target gene activation [22,23,24]. The human core complex of NuRF has similar properties to its *Drosophila* counterpart, and shares the orthologs of three of four components, BPTF (Bromodomain PHD finger transcription factor), SNF2L/SMARCA1 (SWI/SNF-Related, Matrix-Associated, Actin-Dependent Member 1) and the WD repeat-containing protein RBAP46/48. The core BPTF contains a PHD finger and a bromodomain that bind to trimethylated histone H3 lysine 4 (H3K4me3) and acetylated histones, respectively [25]. The less-characterized BAP18 (BPTF-associated protein of 18 kDa), encoded by the chromosome 17 open reading frame 49 (C17orf49) gene, has been identified as a potential interacting partner of both NuRF and mixed-lineage leukemia protein 1 (MLL1) chromatin complexes [20,26], suggesting that BAP18 could link active chromatin reader and writer activities (reviewed in [27]). Indeed, H3K4me3 is highly enriched at transcription start sites (TSSs) of active genes and controls gene transcription [28,29]. Promoters at critical genomic positions, such as active chromatin domains, offer multiple looping possibilities and thus regulate neighboring transcriptional units.

Our study revealed that HMGXB4 targets SB to germinal stem cells, where it enhances transposase expression via chromatin remodeling. The transcriptional activation function is regulated by post-translational modification (e.g., SUMOylation) that controls subcellular trafficking of the protein, resulting in an altered stoichiometry of HMGXB4 in the chromatin remodeling complex(es). We demonstrate that HMGXB4 is an essential but overlooked developmental factor, connecting somatic and germinal stemness in early vertebrate embryogenesis. 

## 2. Results

### 2.1. HMGXB4 Is Regulated by SUMOylation 

Despite its potential role in Wnt signaling regulation [18,19], the molecular function of HMGXB4 is relatively uncharacterized. To clarify its role, we commenced our study to determine the interacting partners of the HMGXB4 protein. To this end, we performed a yeast two-hybrid (Y2H) assay using a human HeLa cDNA library (Method, Appendix A). This screen identified SUMO1 and PIAS1 (confirmed by co-IP (Figure 1A and Appendix A), suggesting that HMGXB4 is likely modulated post-translationally by the components of the SUMOylation machinery. To find out if HMGXB4 is covalently modified by SUMO1 [30], a tagged HMGXB4-HA (either human or zebrafish origin) was co-expressed with SUMO1, and the protein extracts were analyzed by Western blotting (Figure 1A and Appendix A). The slower migrating band was identified as a SUMOylated protein product (Figure 1A), suggesting that SUMO1 modifies HMGXB4 via a covalent bond formation to diglycine. While all three SUMO versions (e.g., SUMO1,2,3) modified the HMGXB4 protein, SUMO1 was the most potent modifier (Figure 1B).

To map the SUMOylated lysine (K) residues of HMGXB4, we selected those that were phylogenetically conserved among vertebrate orthologs of HMGXB4 (Appendix A and Appendix A) and converted them to arginine (R) by site-specific mutagenesis. In the presence of SUMO1, the combination of K317R and K320R mutations abolished the SUMOylated band(s) (Figure 1C and Appendix A). We used this version, called HMGXB4^SUMO-^, for further experiments. 

SUMOylation is a dynamically reversible process that elicits transient responses controlled by conjugating and deconjugating enzymes. The SUMO moiety can be removed by the SENP (SUMO1/sentrin/SMT3)-specific peptidase) family of SUMO-specific proteases, SENP (1–3) and recycled in a new SUMOylation cycle ([31,32] and reviewed in [33]). Our in vitro SUMOylation assays revealed that both SENP1 and SENP2 reduced SUMO1 conjugation, whereas SENP3 deconjugated SUMO2 (Figure 1D,E). In addition, we found that SUMO1 modification of HMGXB4 was sensitive to the presence of the chemical stress factors, ethanol and H_2_O_2_, suggesting an additional layer of HMGXB4 regulation by stress (Figure 1F). 

SUMOylation can be facilitated by E3 ligases, such as PIAS1 (protein inhibitor of activated STAT 1) [34]. However, our experiments did not support the role of PIAS1 as an E3 ligase for HMGXB4 (Appendix A). Alternatively, PIAS1 might have a transcriptional co-regulation function [35] (not followed up in the current study). 

**Figure 1 ijms-24-07283-f001:**
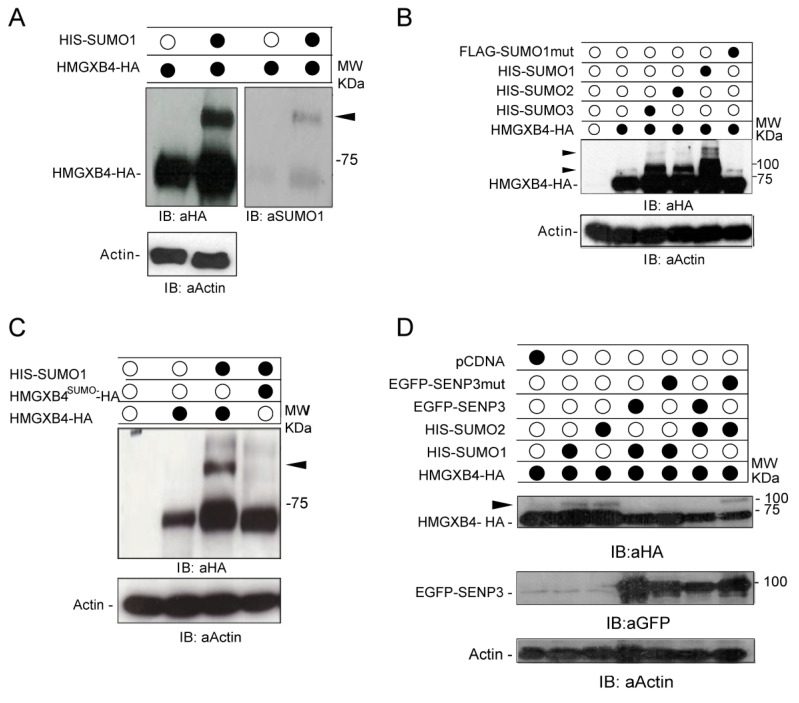
HMGXB4 is regulated by SUMOylation. (**A**) (Left panel) HMGXB4 gets SUMOylated in the presence of SUMO1 (immunoblots). HeLa cells were co-transfected with expression constructs of the tagged versions of the candidate proteins, HMGXB4-HA and HIS-SUMO1. Whole-cell lysates were immunoblotted. A slower migrating band, potentially corresponding to the SUMOylated version of HMGXB4, is marked by a black triangle (Right panel). The membrane is striped and rehybridized using an antibody against SUMO1. (**B**) All three SUMO variants (e.g., 1, 2 and 3) can SUMOylate HMGXB4. HeLa cells were co-transfected with expression plasmids encoding HMGXB4-HA, HIS-SUMO1, 2, 3 or a mutated version of FLAG–SUMO1ΔGG, defective in conjugation. The SUMOylated HMGXB4 versions are expected to appear as slower migrating bands (black triangles) on the immunoblot, using HA-specific antibodies in whole-cell lysates. Importantly, no shifted bands were detectable when HeLa cells were transfected with the mutated version of SUMO1mut, lacking the diglycine C-terminal motif required for its conjugation to substrates SUMO1-∆GG [36]. (**C**) K317R/K320R dual mutations abolish the post-translational modification and SUMOylation of HMGXB4. HeLa cells were co-transfected with expression plasmids encoding HMGXB4-HA, HMGXB4^K317R/K320R^-HA (referred to as HMGXB4^SUMO-^ in the following) and His-SUMO1. The whole-cell HeLa lysates were immunoblotted with an HA-specific antibody. The SUMOylated HMGXB4 is an additional, slower migrating band (marked by a black triangle). (**D**,**E**) SUMOylation of HMGXB4 is reversible by SUMO-specific proteases (SENPs). HeLa cells were co-transfected with expression constructs of HMGXB4-HA, HIS-SUMO1 and SENP1, SENP2 and SENP3 or their SUMO deconjugation defective mutant versions (mut), SENP1^602S^, SENP2^C547S^ and SENP3^C532S^ [32,37], respectively. Whole-cell lysates were subjected to immunoblotting with anti-HA to monitor HMGXB4 and FLAG/GFP-specific antibodies to detect the SENP proteins. The SUMOylated HMGXB4 appears as an additional and slower migrating band (marked by black triangle). (**F**) SUMOylation of HMGXB4 is stress sensitive. The in vitro SUMOylation assay was performed under various cellular stress conditions. Note that ethanol and H_2_O_2_ treatments enhanced the abundance of the SUMOylated bands compared to the untreated condition. The SUMOylated HMGXB4 appears as an additional and slower migrating band (marked by black triangle).

### 2.2. HMGXB4 Is Associated with Nucleosome Remodeling Activities Regulated by Subcellular Trafficking

SUMOylation might affect several aspects of the target protein, including structure, interaction partners, cellular localization, enzymatic activity or stability (reviewed in [33]). We detected both HMGXB4^wt^ and HMGXB4^SUMO-^ at similar levels at different time points following a cycloheximide treatment (Appendix A), indicating that SUMOylation did not affect the stability of HMGXB4 protein. Alternative to a hypothesis-driven strategy, we have performed an unbiased high throughput protein interactome analysis to decipher the effect of SUMOylation on HBGXB4 function. To this end, we used a triple SILAC pull-down approach, suitable for the relative quantification of proteins by mass spectrometry [38] (Appendix A and Appendix A). To find out which functions of HMGXB4 are affected by the presence of the transposase, we also transfected HEK-293T cells with HA-tagged HMGXB4^wt^ and HMGXB4^SUMO-^ in the presence/absence of SB transposase and/or SUMO1 (Appendix A). 

According to gene ontology (GO) analysis, the interactome of HMGXB4 could be characterized by the top terms of *Wnt signaling, Ribonucleoprotein complex, Structural and cytoskeletal, Host response to viruses, Transcriptional regulation, Translation initiation, Nonsense-mediated decay (NMD) and Regulation of metabolism*. 

Among the differentially recruited proteins of HMGXB4^SUMO-^, we detected C17orf49 (alias BAP18) (Figure 2A). Since the interaction of (wild-type) HMGXB4 and C17orf49/BAP18 has been reported previously [39,40] and [41], we performed an additional co-IP experiment with HMGXB4^WT^ and HMGXB4^SUMO-^. Our co-IP revealed significant interactions with both HMGXB4^WT^ and HMGXB4^SUMO-^ with HMGXB4^SUMO-^, capturing more C17orf49/BAP18 protein molecules (Figure 2B). Since HMGXB4 and C17orf49/BAP18 are considered members of the ATP-dependent chromatin reading complex (NuRF), involved in reading active marks at H3K4 [20], the observed differential binding raised the possibility that SUMOylation might control the participation of HMGXB4 in chromatin remodeling activities. 

Regardless of the SB transposase presence, the top GO categories of the HMGXB4 interactome were similar (Appendix A); however, the affinity of the interactions appeared stronger and/or the higher number of interacting partners in most of the shared GO categories (Figure 2A–C and Appendix A), suggesting that the transposase modulates these functions of HMGXB4 via intensifying interactions to its partners. 

The C17orf49/BAP18 in the HMGXB4 interactome could be noteworthy because a C17orf49/BAP18 can co-complex with the NuRF and SET1/MLL complexes [27,39], thus might provide a potential link between active chromatin reader and writer activities (reviewed in [27] (Appendix A)). Coordinated chromatin reading and writing activities might explain the significance of HMGXB4 as a transcription activator. The interactome data revealed that SUMOylation affected the affinity of HMGXB4 to its chromatin remodeling partner, suggesting that this post-translational modification might also affect its transcriptional modulatory function on SB transposition.

To address this possibility, we performed both transcription and SB transposition assays. In the transient luciferase reporter assays, 5′UTR-luciferase, HMGXB4^WT^, HMGXB4^SUMO-^, SUMO1 and PIAS1 were co-transfected into HeLa cells in various combinations. We observed that while HMGXB4^WT^ and HMGXB4^SUMO-^ could transactivate reporter expression from the left IR of the SB transposon, overdosing SUMO1 had attenuated the transcriptional activation of HMGXB4^WT^ (Figure 2D). Accordingly, the HMGXB4-mediated enhancement of the transposition [10] was also mitigated in the presence of co-transfected SUMO1 (Figure 2E). In line with the in vitro SUMOylation experiments (Appendix A), elevated PIAS1 levels had no obvious consequence on SB activities in either of the assays (Figure 2E). Altogether, and according to the interactome data, SUMOylation compromises the transcriptional activator function of HMGXB4.

In the differential proteome of HMGXB4^WT^/HMGXB4^SUMO-^, we observed a group of proteins associated with nucleolar functions (Figure 3A), suggesting that the SUMOylation potentially affects nucleolar activities and/or subcellular trafficking. The nucleolar interactors, by contrast to C17orf49/BAP18 (NuRF/MLL complexes), had higher affinity to HMGXB4^WT^ versus HMGXB4^SUMO-^, and were associated with the GO categories of *Translation initiation and elongation, Transcriptional control*, *Non-sense-mediated decay* and *Ribonucleoprotein complex (Ribosomal structure*). The presence of the SB transposase intensified the affinity of interaction in all of these GO categories (Figure 3B,C and Appendix A), suggesting that via HMGXB4, the transposase might sponge on transcription activation, non-sense-mediated decay, transcript processing and protein translation machineries of the host cell. 

To demonstrate that SUMOylation affects the subcellular trafficking of HMGXB4, we used confocal microscopy. We co-transfected expression vectors of HA-tagged HMGXB4^WT^, HMGXB4^SUMO-^, EGFP-tagged SUMO1 (EGFP-SUMO1) and SB (EGFP-SB) into HeLa cells in various combinations, and subjected the cells to microscopy. This strategy revealed a clear antagonistic subcellular localization pattern of HMGXB4^WT^ and HMGXB4^SUMO-^. HMGXB4^SUMO-^ stayed in the nucleoplasm (even in excess of SUMO1), whereas HMGXB4^WT^ co-localized with the nucleolar marker fibrillarin (Figure 3D and Appendix A). These observations support that SUMOylation induces nucleolar compartmentalization of HMGXB4. 

Subnuclear trafficking of HMGXB4 affects the localization of the SB transposase (Figure 3C and Appendix A), which translocates from the nucleolus to the nucleus in a manner similar to HMGXB4^SUMO-^, suggesting that SB transposition may piggyback the nuclear function of HMGXB4, involved in transcription initiation. Curiously, unlike HMGXB4, the SB transposase was enriched in the perinuclear nuage of cells (Figure 3C), where the machinery of piRNA biogenesis is located [42]. 

Collectively, the transcriptional activation function of HMGXB4 is modulated by post-translational modification. SUMOylation regulates subnuclear trafficking, resulting in the accumulation of HMGXB4 in the nucleolus and, thus, its physical sequestration from the chromatin remodeling complexes.

### 2.3. HMGXB4 Is among the Earliest Genes Expressed in Vertebrate Development

The above data suggest that it could transactivate transposase expression when HMGXB4 is present in the nucleus. What is the expression profile of HMGXB4 in the host organism? To address this, we followed the line that HMGXB4 was reported to regulate the Wnt signaling pathway [18,19]. As the Wnt signaling pathway is one of the earliest cellular processes activated in the developing embryo (reviewed in [43]), we determined the expression profile of HMGXB4 during early development. 

First, we monitored the expression of zHMGXB4 during zebrafish development from the zygote stage to hatching. Our qPCR data show that zHMGXB4 is highly expressed already in the zebrafish zygote, and its expression level drops from the unfertilized oocyte to the zygotic transition (Figure 4A). Following its sharp decline after the blastula stage, zHMGXB4 expression is detectable again in the pharyngula (Figure 4A). It can therefore be assumed that HMGXB4 acts not only in the first steps of embryogenesis, but also at the later stages of embryonic development. 

We also analyzed mammalian data. Our analysis of single-cell transcriptome datasets (scRNA-seq) of mouse and human preimplantation embryos [45,46] revealed that HMGXB4 is among the ~300–400 genes detected at a significant level (Log2 FPKM > 2) in every cell of both species (Figure 4B,C). Furthermore, HMGXB4 is highly expressed prior to embryonic gene activation (EGA), followed by a reduced (but still significant) expression in the preimplantation embryo (Figure 4D). 

The above analyses revealed that the maternally expressed HMGXB4 continues to be expressed during early development, but its level declines sharply after the blastula stage. The observed expression fluctuations during development accord with its proposed regulatory function of the Wnt signaling [18,19]. Importantly, the expression pattern of HMGXB4 is phylogenetically conserved in vertebrates.

### 2.4. HMGXB4 Is Part of a Regulatory Network of Stemness 

To gain insight into the transcriptional regulation of HMGXB4 during early embryogenesis, we performed an integrative analysis of RNA-seq (N~300), Hi-C, ChiP-seq/ChIP-exo/CUT&RUN of transcription factors (TFs) and histone modification data over the HMGXB4 locus in HeLa, embryonic stem cells H1_ESCs and human early embryos [44,46,47,48]. Our analysis uncovered that HMGXB4 expression might be activated by both MAPK1 (alias ERK2) and ELK1 transcription factors (Figure 4E), implicated in coordinating pluripotency and self-renewal pathways [44]. Following the line that HMGXB4 is a host-encoded factor captured by a transposable element (TE), we asked whether the KRAB-ZNF/TRIM28 epigenetic repression system, playing an essential regulatory role in suppressing TE-derived transcription [49] might also be involved in controlling HMGXB4 expression. The KRAB-ZNF/TRIM28 system utilizes KRAB-ZNF transcriptional regulators to target the TRIM28/KAP1-mediated transcriptional repression machinery to specific genomic locations in higher vertebrates [50]. Thus, we data mined ChIP-exo seq peaks of 230 KRAB-ZNF proteins, reported from HEK293 and human embryonic stem cells [51]. This approach identified repressive KRAB-ZNF proteins (e.g., ZNF468, ZNF763 and ZNF846), harboring significant peaks (adjusted *p* < 1 × 10^−7^, compared with total input control) at the transcription start site (TSS) of HMGXB4 (Figure 4F,G). Does the expression of the HMGXB4 match the transcription dynamics of the potential regulators during early embryogenesis [46]? This analysis supported that the expression of HMGXB4 accords with the dynamic expression of MAPK1/ERK2 throughout the human preimplantation embryogenesis, whereas it was antagonistic to that of ZNF468 (Figure 4H). The above data supported that HMGXB4 expression might be activated by pluripotency/self-renewal transcription factors (e.g., ERK2, ELK1), whereas epigenetically controlled by repressive histone marks deposited by the KRAB-ZNF/TRIM28 system (targeted by ZNF468). 

The analysis of the three-dimensional (3D) conformation of human ESC genomes revealed that the promoter of HMGXB4 is located near the boundary of active chromatin compartments (Figure 5A), which could enable multiple interactions between enhancers and promoters in stem cells. This boundary is marked by CTCF (Figure 5A) and co-occupied by ChIP-seq peaks for H3K27ac, MED1 (Mediator 1), POU5F1/OCT4 and POLII (adjusted *p* < 0.01, BH corrections) over the TSS of HMGXB4, connecting gene expression and chromatin architecture [52]. Adding additional layers of CUT&RUN data analysis for H3K4me3 and H3K27me3 uncovers that the TSS of HMGXB4 has an enrichment for H3K4Me3 (adjusted *p* < 0.01, BH corrections) but not for H3K27Me3 in human germinal vesicle (GV) stage oocytes, four-cell, eight-cell and ICM (inner cell mass). The presence of H3K4Me3 over the HMGXB4 promoter in oocytes, two-cell and four-cell stages suggests that its chromatin architecture gets activated even before the embryonic genome activation (EGA).

### 2.5. HMGXB4 Links Pluripotent and Germinal Stem Cells 

Detecting a high expression signal in oocytes motivated us to analyze single-cell transcriptome datasets of sex-specific germ cells (GSE63818) [53,54]. We readily observed elevated HMGXB4 expression in female and male germ cells, compared with somatic cells in the same niche (Figure 5B). In addition, we analyzed scRNA-seq data of germ cells upon differentiation from pluripotent stem cells in vitro (GSE102943) [55]. This analysis revealed that HMGXB4 was expressed at comparable levels in both pluripotent and germinal stem cells (CD38^+^) and that the expression of HMGXB4 was maintained during the pluripotent to germ cell transition (Figure 5C). Its expression, by contrast, declined in differentiated germ cells (CD38^-^) (Figure 5C), suggesting that HMGXB4 expression is specific to stem cells. This pattern of HMGXB4 expression is supported by the analysis of a large cohort of scRNA-seq datasets of gonad development (~3000 single cells, GSE86146) [53] (Figure 5D), identifying HMGXB4 as a novel factor specific for the stem cell lineages in the germ line. 

To substantiate the differential expression of HMGXB4 between stem versus differentiated germ cells, we used a mammalian (rat) spermatogonial stem cell (SSC) differentiation model. These SSCs maintain their stemness on mouse embryonic fibroblast (MEF) feeders and differentiate when MEFs are replaced by STO (SNL 76/7) cells [56]. In conjunction with the single-cell transcriptome analyses, this approach supported the specific expression of HMGXB4 in spermatogonial stem cells. In contrast, its expression levels (both transcript and protein) sharply dropped upon differentiation (Figure 6A,B), indicating that the expression of HMGXB4 is tightly regulated between self-renewing and differentiated states. 

**Figure 5 ijms-24-07283-f005:**
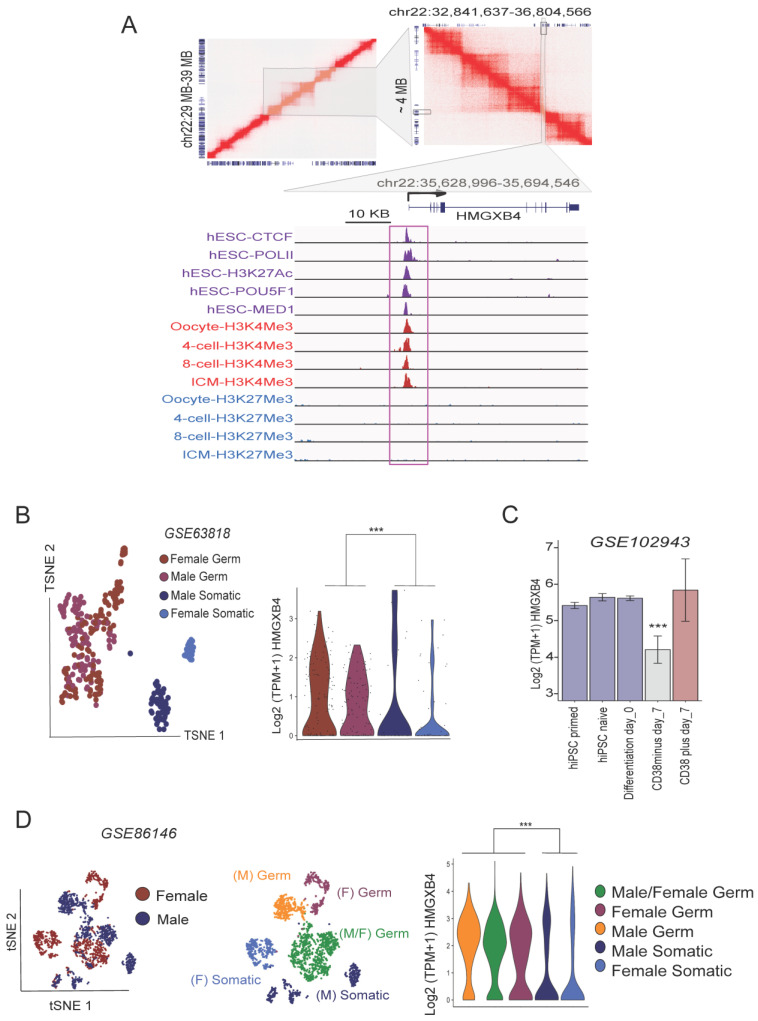
HMGXB4 Connects Pluripotent and Germinal stemness. (**A**) Upper panels: Pairwise contact matrices inferred from Hi-C data generated from human embryonic stem cells [47] (GSE116862) show a region between 29–39 MB on chromosome 22 at 5 kb resolution (left). The intensity of each pixel represents the normalized intensity of observed contacts between a pair of loci. The intensity of red color is proportional to the intensity of contact between two loci plotted on the X and Y axes. Gene models corresponding to these loci are placed on the bottom and left side of the map. Neighbor loops encompassing ~4 MB DNA sequence are further zoomed (right, gray shaded region). HMGXB4 gene is situated at the boundary of the interacting loop (boxed), which is further zoomed (lower panel, gray shaded region): Integrative genome visualization of the active loop boundary around the HMGXB4 locus. Binding profiles of ChIP-seq peaks (boxed) for CTCF, POLII, H3K27Ac, POU5F1 and MED1 (purple peaks) (GSE69646) [57] over the TSS of HMGXB4 (arrow). CUT&RUN profiling for H3K4me3 (dark red peaks) and H3K27me3 (blue peaks) in human germinal vesicle (GV) oocytes, 4-cell, 8-cell and Inner cell mass (ICM) (GSE124718) [48] at the TSS of HMGXB4. Note that no significant H3K27me3 peaks were detected on the shown locus. (**B**) TSNE plot illustrates the single cell clusters from the development of male and female human Primordial Germ (hPG)-like cells (GSE63818) based on the expression of Most Variable Genes (MVGs). The single-cell clusters are distinguishable as Male/Female Germ/Somatic cells. (left panel). Violin plots on the right panel display the Log2 normalized expression of HMGXB4 in the clusters using the same color codes. Every dot represents a single cell. Note the germ stem cell-specific expression of HMGXB4. (**C**) HMGXB4 is expressed throughout the differentiation of human induced pluripotent stem cells (iPSCs) to CD38^positive^ primordial germ (hPG)-like cells, marked by CD38 (GSE102943). Barplots show the transcriptomic changes of HMGXB4 during the differentiation (Diff) process. Note the downregulation of HMGXB4 in CD38^minus^ (somatic) cells, whereas HMGXB4 expression was not significantly altered in hiPSCs, and CD38plus day_7. *** *p*-value < 0.05, *t*-test of HMGXB4 expression in CD38minus day_7, compared with all shown conditions. (**D**) TSNE plot illustrates the clusters of male and female human fetal germ single cells (upper panel). The bottom panel shows the clusters defined by the topmost variable gene expression. The clusters were annotated using published transcriptional markers as male and/or female germ/somatic cells (GSE86146). Violin plots (lower panel) display the Log2 normalized expression of HMGXB4 in the various clusters (color code is the same as on the middle panel). Every dot represents a single cell.

### 2.6. HMGXB4 Activates Sleeping Beauty Transposition in the Germline 

Specific expression of HMGXB4 in the germline tempted us to ask whether HMGXB4 is a host factor that potentiates SB transposition in the germline. To answer, we established a quantitative SB transposon excision assay in SSCs, cultured on MEF or STO cells (Figure 6C,D). In the assay, SB transposase expression is driven by the transposon’s 5′UTR that contains the HMGXB4 target sequences required for the activated transcription. Our assay revealed that the frequency of SB excision was high in SSCs kept on MEFs, while sharply declined upon culturing on STO cells, which triggers differentiation (Figure 6D). Thus, the rate of transposon excision matches the expression level of HMGXB4, suggesting that HMGXB4 is an essential host-encoded factor for efficient SB transposition in germinal stem cells. Notably, at a decreased level, SB excision still occurs in differentiated cells (Figure 6D), agreeing with the assumption that the requirement of HMGXB4 for transposition is not absolute [10]. 

### 2.7. The Transcriptional Activation Activity of HMGXB4 Is Conserved in Vertebrates 

A returning question of transposon-host interaction studies concerns their cross-species conservation. While the HMGXB4 gene exists in all vertebrate species, the coding sequence of the fish version is significantly divergent (35%) from its human counterpart (Appendix A). As SB is originated from fish genomes [11], we asked if the transcriptional enhancer effect [10] of the human hHMGXB4 on SB transposition was reproducible in (zebra)fish embryos. In a reporter assay, luciferase expression was controlled by the 5′-UTR region of the transposon or by a mutated version, where the HMGXB4-responding region was deleted (pLIRΔHRR) [10]. Luciferase activity driven by the 5′-UTR was detectable in zebrafish (*Danio rerio*) embryo extract and depended on the presence of the HMGXB4 responding region (Figure 6E). In addition, we transiently overexpressed zebrafish zHMGXB4 or hHMGXB4 by co-injecting the corresponding expression constructs with the reporter into zebrafish embryos. The presence of zHMGXB4 elevated the transcription of the 5′UTR-luciferase reporter three-fold above the level obtained using hHMGXB4 in a similar assay (Figure 6F). The activator effect of zHMGXB4 was even higher than the human ortholog when tested in a colony-forming transposition assay performed in Hela cells (Figure 6G). While SB transposition responded more robustly to zHMGXB4 than hHMGXB4, the effect/pattern was similar, confirming our hypothesis that HMGXB4 is a conserved host factor of SB transposition in vertebrates [17]. Thus, the transcriptional activation function of HMGXB4 can be modeled by SB transposition from fish to human cells.

## 3. Discussion

Viruses and transposons frequently piggyback the “essential” cellular mechanism(s) of the host. The role of HMGXB4 in *Sleeping Beauty* (SB) transposition is conserved from fish to humans, supporting the assumption that HMGXB4-SB transposon interaction can be generally modeled in vertebrates [17]. Our study identifies HMGXB4 as a novel factor linking pluripotency to the germline in vertebrates and as a host-encoded factor that targets SB transposition to germinal stem cells. 

The maternally expressed HMGXB4 is among the first set of expressed genes in the embryo, and in agreement with its regulatory role in Wnt signaling [18,19], its expression level is dynamically changing throughout embryogenesis. HMGXB4 is activated by its promoter, responding to the parental-to-zygotic transition, marks stemness and maintains its expression during germ cell specification. The promoter of HMGXB4 is located at an active chromatin domain boundary, offering multiple looping possibilities with neighboring genomic regions in stem cells. Thus, besides the germline, the recruitment of HMGXB4 supports efficient SB transposition during early embryogenesis in various somatic progenitor cells (including therapeutically relevant cell types) [58]. 

Via HMGXB4, SB might piggyback multiprotein complexes, capable of depositing and reading active chromatin marks at H3K4. Notably, C17orf49/BAP18 provides a physical bridge between NuRF and MLL complexes [27], thereby linking histone H3K4 methyltransferase- and ATP-dependent nucleosome remodeling activities. Via ERK2/MAPK1-ELK1, MED1, CTCF and POU5F1, HMGXB4 is part of the transcription regulatory network, implicated in regulating differentiation potential and self-renewal [44]. 

A reversible post-translational modification, SUMOylation, regulates the translated HBGXB4. While SUMOylation does not affect the stability of the HMGXB4 protein, it regulates its binding affinity to its protein-interacting partners. The ability to associate with chromatin remodeling complexes enables HMGXB4 to act as a transcriptional activator, whereas SUMOylation serves as a signal for its nucleolar partition. Trafficking of HMGXB4 between nucleoplasm and nucleolus affects the stoichiometry of the HMGXB4-containing protein complexes and provides flexible regulation of the transcription-activating function. The nucleolar functions of HMGXB4 include pre-mRNA processing and ribosome biogenesis. 

SB piggybacks both SUMOylated and non-SUMOylated functions of HMGXB4. The SB transposase follows HMGXB4 during its subnuclear trafficking to the nucleolus, which might promote the processing of SB’s enforced de novo transcripts. Notably, like SB, viruses are reported to target the nucleolus to favor viral transcription, translation and/or modulate cell cycle to promote viral production (reviewed in [59]). In addition, the SB transposase is also enriched in the perinuclear nuage-like structure. This structure was associated with piRNAs, known to repress transposable elements via RNAi (reviewed in [42]). Although SB is not endogenous in humans, SB might be capable of recognizing an evolutionarily conserved feature of piRNA biogenesis. 

Unlike retrotransposons that rarely mobilize in undifferentiated germinal stem cells [3], SB targets this cell type. The HMGXB4-mediated germline targeting is a likely conserved feature of the *Tc1-like* family of transposons (where SB belongs) in vertebrates. Whilst the *Drosophila P* element utilizes a similar strategy to target germinal stem cells [2], the host-encoded targeting factor is yet to be identified and could not be identical to the vertebrate-specific HMGXB4. SB transposition, by contrast to the *P* element [60], is not restricted to the germline and can efficiently transpose in various somatic (especially stem) cells [12,58]. The significance of this observation is not entirely understood. The ability to transpose in many somatic stem cells might be a simple “side effect” of recruiting a host-encoded factor associated with Wnt signaling. By a less understood mechanism, besides the germline, LINE-1 retrotransposition can also get activated in somatic (e.g., neuronal and placental) progenitors [61,62,63]. 

While HMGXB4 is involved in the epigenetic regulation of gene expression, it is controlled by the KRAB-ZNF/TRIM28-mediated epigenetic repression mechanism [64]. One of the key functions of the KRAB-ZNF/TRIM28 system is to suppress the TE-derived transcription [49]. In an evolutionary battle, some TEs establish strategies to modulate/escape host-encoded suppression mechanisms, raising a potential scenario of a regulatory loop when SB interferes with the suppression of HMGXB4 expression.

Notably, the SUMO-specific deconjugation process of HMGXB4 is stress-inducible and thus could activate transcription upon environmental changes. The stress sensitiveness of HMGXB4 (this study and [65]) would enable SB transposon to sense and react to cellular stress, a known feature of transposable elements [66]. Similarly, a stress-responsive SUMO-regulated chromatin modification has been implicated in reactivating integrated viruses in the genome (e.g., heterochromatin histone demethylase, JMJD2A in Kaposi’s sarcoma-associated herpes virus (KSVH)) [67]. 

Deciphering a relationship between a host-encoded factor piggybacked by a transposable element also spotlights the poorly characterized developmental gene HMGXB4. Although HMGXB4 has been suggested to be a member of the NuRF complex [20], it is unlikely to be a core member. Notably, the activity of HMGXB4 is stem/progenitor cell-specific, and the expression level of HMGXB4 drops sharply upon differentiation and stays at an undetectable level in differentiated cells. Importantly, however, HMGXB4 is epigenetically regulated and stress-sensitive, and when expressed, it can co-complex with chromatin remodeling complexes. HMGBX4 might activate transcription and mRNA processing in any cell type, thus providing the regulatory machinery of target gene activation and production. Following this line, aberrant activation of HMGXB4 in differentiated cells (e.g., cancer) might result in undesirable transactivation of target gene expression. In this context, HMGXB4 has been identified as a target of splicing regulatory proteins upon an epithelial–mesenchymal transition (EMT) [68] and implicated in playing a role in the inflammatory response [65], suggesting that HMGXB4 could be a so far-overlooked factor of human pathologies. Overall, our study highlights HMGXB4 as an evolutionarily conserved host-encoded factor that assists *Tc1/Mariner* transposons to target the germline, which was necessary for their fixation and may explain their abundance in vertebrate genomes.

## 4. Materials and Methods

### 4.1. Constructs

pCAG-Venus-SB10, the SB10 transposase [11] gene, was cloned into pCAG-Venus. The zHMGXB4 coding sequence was PCR amplified from the cDNA of zebrafish embryo with zHMG EcoRI Fwd. and NotI Rev. primers. The PCR product was digested with EcoRI and NotI restriction enzymes and subcloned into the corresponding sites of the pcDNA3.1 vector; HMGXB4-HA, C-terminal HA (hemagglutinin peptide, YPYDVPDYA)-tagged versions of the HMGXB4 were obtained by inserting HA tag downstream of the coding region of the zebrafish/human HMGXB4 protein by PCR and cloning into the HindIII/XhoI sites of pcDNA3.1.

### 4.2. Mitotic Inactivation of MEFs

Mouse embryonic fibroblasts (MEFs) are often used as feeder cells in embryonic stem cell research. MEFs were isolated from 12.5 to 13.5 post coitum (p.c.) mouse embryos. The embryos were dissociated and then trypsinized to produce single-cell suspensions. After expansion, confluent MEFs cells were treated with 10 μg/mL mitomycin-C (Merck Limited, Merck KGaA, Darmstadt, Germany) for 2 h in DMEM at 37 °C. Cells were washed twice with PBS, followed by trypsinization, and counted before dilution and plating.

### 4.3. Rat Spermatogonial Stem Cells Culturing

Rat spermatogonial stem cell lines were cultured on mitomycin-C treated MEFs in spermatogonial culture medium (SG medium) as described [69]. The cells were passaged at 1:3 dilutions onto a fresh monolayer of MEFs every 10–14 days at 3 × 10^4^ cells/cm^2^. For passaging, cultures were first harvested by gently pipetting them free from the MEFs. After harvesting, the clusters of spermatogonia were dissociated by gentle trituration with 20–30 strokes through a p1000 pipette in their SG culture medium. The dissociated cells were pelleted at 200× *g* for 4 min, and the number of cells recovered during each passage was determined by counting them on a hemocytometer. Feeder Removal Micro Beads (Miltenyi Biotec, Bergisch-Gladbach, Germany) were used to deplete MEFs while co-culturing with rat spermatogonial stem cells for gene expression and protein studies. The protocol for co-culturing the rat spermatogonial stem cells with STO cells was performed as in [56]. The separation of STO cells from rat spermatogonial stem cells was carried out using magnetic-activated cell sorting (MACS) from (Miltenyi Biotec, Bergisch-Gladbach, Germany). 

### 4.4. Microinjection of Zebrafish Embryos

To measure the promoter activity of the left inverted repeat (LIR) of the *Sleeping Beauty* transposon in zebrafish embryos, a microinjection mix, containing 50 ng/μL of firefly luciferase and 0.2 ng/μL Renilla reporter plasmids containing Buffer Tango 10× to a final concentration of 0.5×, and phenol red solution to a final concentration of 0.05% were injected into fertilized eggs of wild-type *Danio rerio*. The injected embryos were incubated for 24–48 h at 28.5 °C in egg water. For measurement of promoter activity, the egg water was removed. The embryos were washed with PBS, followed by lysis with 50 μL of passive lysis buffer (PLB) 1× for 30 min at room temperature, shaking at 150 rpm. Firefly and Renilla luciferase activities were measured according to the manufacturer protocol (Dual-Luciferase Reporter Assay System, Promega GmbH, Darmstadt, Germany).

### 4.5. Transient Expression of Proteins in HeLa or HEK293 Cells

HeLa or HEK293 EBNA cells were cultured in DMEM medium supplemented with L-glutamine, penicillin/streptomycin and 10% FBS (Thermo Fischer Scientific, Waltham, MA, USA). Cells were transiently transfected at 50–65% confluency with QIAGEN-purified plasmid DNA using JetPEI or FuGENE transfection reagent according to the manufacturer’s instructions. After 48 h post-transfection, cells were lysed in RIPA lysis buffer containing 25 mM Tris-HCl pH 7.6, 150 mM NaCl, 1% NP-40, 1% sodium deoxycholate and 0.1% SDS supplemented with protease cocktail inhibitors (Roche Holding AG, Basel, Switzerland) and subjected to Western blot analysis.

### 4.6. Total Protein Quantification

Protein concentrations were measured by using a calorimetric technique at a wavelength of 562 nm (OD562) by bicinchoninic acid assay (Pierce™ BCA™ Protein-Assay- Roche Holding AG, Basel, Switzerland). Samples containing known concentrations of bovine serum albumin (BSA) were used as a standard.

### 4.7. SDS-Polyacrylamide Gel Electrophoresis

Proteins were separated by their molecular weight using SDS-polyacrylamide (SDS–PAGE) gels ranging from 10–15%. The total protein (50 μg) lysate was mixed with 1× Laemmli buffer incubated for 5 min at 95 °C and resolved at 80 V in SDS–PAGE running buffer (50 mM Tris-HCl, 196 mM glycine, 0.1% SDS, pH 8.4). After electrophoresis, the gels were subjected to Western blotting.

### 4.8. Western Blotting

Cell or tissue extracts were resolved by 7.5–10% SDS-polyacrylamide gel electrophoresis (SDS–PAGE) and electro-transferred to polyvinylidene difluoride (PVDF) membranes (GE Healthcare, Chicago, IL, USA). The membranes were incubated with 5% nonfat dry milk at room temperature for 1 h and probed overnight with specific antibodies at 4 °C. The immune complexes were detected by enhanced chemiluminescence (ECL, Pierce) with anti-mouse, anti-goat, anti-rat, or anti-rabbit immunoglobulin G (IgG)-coupled horseradish peroxidase (Pierce) as the secondary antibody.

### 4.9. Co-Immunoprecipitation

HeLa or HEK293 cells (2 × 10^6^) were co-transfected with indicated plasmids. The whole-cell lysate was prepared using extraction buffer (Tris-HCl 50 mM at pH 7.4, NaCl 150 mM, EDTA 1 mM, NP-40 1% and Na-deoxycholate 0.25%) supplemented with protease inhibitor cocktail (Roche, Mannheim, Germany). For immunoprecipitations, equal amounts of the lysate (containing 500 μg of total cellular protein from HeLa cells; 2 × 10^6^) were precleared with protein G-agarose beads (Sigma, St. Louis, MO, USA). Precleared extracts were incubated with 1 μg rat monoclonal anti-HA (Roche) for 2 h at 4 °C. Precipitates were washed extensively in lysis buffer; bound complexes were eluted with 2× SDS–PAGE sample buffer and resolved by 7.5–10% SDS–PAGE. Immunoblotting was performed according to standard procedures and proteins were detected with the indicated antibodies. Total protein lysates were prepared in 1× SDS–PAGE sample buffer and 10 μg of proteins were separated by 7.5–10% SDS–PAGE. Antibodies were detected by chemiluminescence using ECL Advance Western Blotting Detection Kit (GE Healthcare, Chicago, IL, USA).

### 4.10. SUMOylation Assay

HeLa cells (2 × 10^6^) were co-transfected with the expression constructs in 10 cm dishes with 2 μg each of His-SUMO1 and HA-tagged wild-type HMGXB4 or mutants. At 48 h post-transfection, cells were lysed by radioimmunoprecipitation assay (RIPA) buffer containing 10 mM *N*-ethylmaleimide (NEM) and protease inhibitors. For immunoprecipitation, equal amounts of lysates (containing 5 mg of total cellular protein from HeLa cells) were precleared with protein G-agarose beads (Sigma, St. Louis, MO, USA). Precleared extracts were incubated with 1 μg rat monoclonal anti-HA (Roche, Mannheim, Germany) for 2 h at 4 °C. Precipitates were washed extensively in lysis buffer; bound complexes were eluted with 2 × SDS–PAGE sample buffer and resolved by 7.5–10% SDS–PAGE. Immunoblotting was performed according to standard procedures, and proteins were detected with the anti-HA antibodies. Antibodies were detected by chemiluminescence using ECL Advance Western Blotting Detection Kit (GE Healthcare, Chicago, IL, USA).

### 4.11. Protein Stability Assay

HeLa cells (3 × 10^6^) were co-transfected with the expression constructs in 10 cm dishes with 5 μg each of His-SUMO1 and HA-tagged wild-type HMGXB4 or mutants. After 12 h transfection, cells were incubated with 100 µM cycloheximide (Sigma) for 0 to 72 h and then harvested with radioimmunoprecipitation assay (RIPA) buffer containing 10 mM *N*-ethylmaleimide (NEM) and protease inhibitors. Equal amounts of total proteins from each treatment were taken to perform Western blot analysis.

### 4.12. Stable Isotope Labeling with Amino Acids in Cell Culture (SILAC)

This protocol relies on the incorporation of amino acids containing substituted stable isotopic nuclei (e.g., ^12^C, ^13^C and ^13^C/^15^N) into proteins in living cells. The three cell populations are grown in culture media that are identical except that one medium contains a “Light,” and the other two media a “Medium Heavy” (or Medium) and “Heavy”, form of a particular amino acid (^12^C-Arginine, ^13^C-Arginine and ^15^N-Arginine, respectively). The mass spectra data were analyzed using MetaCore from GeneGo Inc. St. Joseph, MI, USA (www.genego.com). A fold-change cutoff of 0.5, with *p* < 0.05, was set to identify proteins whose expression was significantly differentially regulated. Enrichment analysis was conducted using GeneGo curated ontologies and Gene Ontology to provide a quantitative analysis of the most relevant biological functions represented by the data (Appendix A).

### 4.13. Mass Spectrometry

A triple SILAC pull-down experiment using anti-HA resin to investigate interaction partners of HMGXB4 and HMGXB4 defective SUMOylation in the presence/absence of *Sleeping Beauty* in transiently transfected HEK293T cells. The cells were cultured in SILAC DMEM, High Glucose (4.5 g/L), w/o L-Arg, L-Lys, L-Gln (PAA), 10% dialyzed fetal bovine serum (PAA), 4 mM L-glutamine (PAA),1% penicillin/streptomycin (100 IU/100 μg, Invitrogen). For Light Medium, they were supplemented with 28 mg/L L-arginine ^12^C_6_·^14^N_4_·HCl (Sigma), 48.7 mg/L L-lysine ^12^C_6_·^14^N_2_·HCl (Sigma), Medium Heavy 28 mg/L L-arginine ^13^C_6_·^14^N_4_·HCl (Sigma), 48.7 mg/L L-lysine ^13^C_6_·^14^N_2_·4,4,5,5-D_4_·HCl (Sigma), Heavy Medium 28 mg/L L-arginine ^13^C_6_·^15^N_4_·HCl (Sigma) and 48.7 mg/L L-lysine ^13^C_6_·^15^N_2_·HCl (Sigma). The cells were cultured for two doublings and were transiently transfected at 80% confluency with 5μg each of pCMV His-SUMO1, pCMV-SB10 and pIRES HA-tagged wild-type HMGXB4 or mutants using JetPEI transfection reagent according to the manufacturer’s instructions. After 24 h transfection, the cells were lysed in lysis buffer containing 25 mM Tris/HCl, pH 7.4 (Carl Roth), 125 mM KCl (Merck), 1 mM MgCl2 (Merck), 1 mM EGTA/KOH pH 8.0 (Carl Roth), 5% glycerol (Merck), 1% NP-40 (Nonidet P 40 Substitute, Sigma), 1mM DTT (Sigma, added freshly) and 1× Protease inhibitor cocktail (Complete, EDTA-free, Roche, added freshly), followed by immunoprecipitation with Anti-HA Agarose beads (Sigma-Aldrich EZview Red Anti-HA Affinity Gel). Mass spectrometry detected the precipitated protein complex for interaction partners, and the results obtained were analyzed by the MaxQuant computational platform.

### 4.14. Quantitative Assay to Monitor Sleeping Beauty Transposon Excision

The transposon and transposase expression constructs were EBNA1-based to avoid fast degradation of the plasmids. pEBNA-SB100X and pEBNA-Tneo are based on the pCEP4/pEBNA vector (a kind gift from T. Willnow, MDC). For the pEBNA-Tneo construct, pEBNA was digested with SnaB1 and Xho1. The Tneo insert was released from the pTNeo construct by Xho1 and Sal1. For the pEBNA—SB100X construct, pEBNA was cleaved by BamH1 and Xho1 and ligated to the insert of UTR-SB100X released with BglII and XhoI from pcDNA3.1 UTR SB100X. In pEBNA—SB100X, the transposase is driven by its promoter located in the left inverted repeat of the transposon (UTR). Following co-transfection of the transposon excision monitoring system (500 ng of each pEBNA-SB100X and pEBNA-Tneo) into SSCs cultured either on MEFs or STO cells, the cells were lysed at the indicated time points. The qPCR detects the precise excision product [13] generated upon SB excision. The PCR primers are listed in Table 1.

The Taqman-excision PCR was performed from cell lysates. A 3 μL volume of the lysate was mixed with primers and probes detecting the excision product and a gene on the input plasmid in a total reaction volume of 20 μL using TaqMan Universal PCR Master Mix ((Thermo Fischer Scientific, Waltham, MA, USA) on the 7900HT Sequence Detection System ((Thermo Fischer Scientific, Waltham, MA, USA). Average Ct values were calculated from quadruplicates of each sample. The amounts of the excision product and input plasmid were calculated using a standard curve for each primer/probe set carried along with each measurement. Standard curves resulted from a dilution row of excision product (pXS plasmid DNA cloned for that purpose) and pEBNA-Tneo input plasmid over a 4–5 logarithmic range. The calculated absolute amounts of excision product were normalized to those of the input plasmid pEBNA-Tneo, and the standard deviation was determined.

Details of an excised pTneo (=pXS) primer/probe binding sites:

ACACAGGAAACAGCTATGACCATGATTACGCCAAGCTTGCATGCCTGCAGGTCGAC*TCTAGAGGATCCCCTACWGTAGGTACCG*AGCTCGAATTCACTGGCCGTCGTTTTACAACGTCGTGACTGGGAAAACCCTGGCGTTACCCAACTTAATCGCCTTGC; underlined: for and rev primer; italic: FAM/TAMRA-probe covering footprint with W = A or T.

### 4.15. Transposition Assay

HeLa cells (5 × 10^5^) seeded in 6-well plates were co-transfected with 90 ng of neo^R^ carrying transposon plasmid plus 150 ng of transposase expression plasmid or mock control along with 90 ng each of His-SUMO1, Flag-PIAS1, HA-tagged wild-type HMGXB4 and HA-tagged HMGXB4 defective of SUMOylation. Two days post-transfection, the cells were trypsinized, and 1/10 of the cells were seeded on 10 cm diameter dishes. Selection was started by culturing the cells in DMEM supplemented with 600 mg/mL G418 (Merck- Millipore, Merck KGaA, Darmstadt, Germany). After 14 days, the selection was terminated by washing the cells with phosphate-buffered saline (PBS), fixed in 10% *v*/*v* formaldehyde, stained with methylene blue in PBS, and counted. The experiments were performed at least thrice, and results are presented as means plus standard deviations.

### 4.16. Dual-Luciferase Reporter Assay

The Luciferase reporter assay was carried out to understand the effect of HMGXB4 on 5′UTR promoter activity of SB in the presence and absence of SUMO1 and PIAS1 proteins. HeLa cells (5 × 10^5^) seeded in 6-well plates were co-transfected with 200 ng of luciferase reporter plasmid along with 150 ng each of His-SUMO1, Flag-PIAS1, HA-tagged wild-type HMGXB4 and HA-tagged HMGXB4 defective of SUMOylation. In all samples, 10 ng of the plasmid pRL-TK (Promega) encoding Renilla luciferase were included for normalization of transfection efficiency. After 48 h, cells were lysed and assayed using the Dual-Luciferase kit (Promega). Relative luciferase activity is the ratio of Firefly to Renilla luciferase activity, normalized to the activity of the reporter alone. The experiments were performed at least thrice, with at least duplicate samples in each study. Results are presented as means plus standard deviations.

### 4.17. Immunofluorescence Staining

Monolayers of HeLa or HeLa-SUMO-1 (8 × 10^4^) cells were grown on coverslips were co-transfected with the expression constructs with 250 ng each of EGFP-SB10, PML-YFP, EGFP-SUMO1 and HMGXB4-HA tag using FuGENE transfection reagent according to the manufacturer instructions. Cells were fixed at 36 h post-transfection with cold 1% paraformaldehyde in Phosphate-buffered saline (PBS) for 20 min at 4 °C and then permeabilized with 1% Tween in PBS and incubated for 2 h at room temperature. Cells were then stained for HMGXB4-HA tag using a HA tag antibody for 2 h in a humid chamber at 37 °C. Covers slips were washed 3× with PBS and stained with Anti-Rat IgG conjugated (Invitrogen) with Cy5 and DAPI for 1 h at 37 °C. The coverslips were removed from the well and rinsed with dH2O to remove excess PBS. Coverslips were placed in Fluorescent Mounting Medium (DAKO cytomation) on a glass microscope slide and dried overnight. The staining was analyzed by confocal microscope (LSM 710 with software ZEN lite 2011; https://www.zeiss.com/microscopy/en/products/software/zeiss-zen-lite.html accessed on 1 December 2022, Carl Zeiss AG, Oberkochen, Germany).

### 4.18. FACS Analysis

Fractions of GFP-positive cells were determined by fluorescence-activated cell sorting using the FACS Calibur (Becton Dickinson, BD Headquarters, Franklin Lakes, NJ, USA). The data were analyzed using CELLQuest v. 3.1 (Becton Dickinson). Briefly, cells were trypsinized in 10 cm tissue culture dishes; the reaction was stopped by adding DMEM medium (Thermo Fischer Scientific, Waltham, MA, USA) containing 10% FCS. Cells were collected in polystyrene tubes and centrifuged at 2500 rpm, 3 min at 4 °C and washed with ice-cold PBS two times. A total of 50,000 cells were analyzed per sample with the same cell flow rates.

### 4.19. SENP Assay

To protect SUMO-conjugated proteins from deSUMOylation, the cell lysates were treated with N-ethylmaleimide (NEM), an inhibitor of SUMO-specific isopeptidases subjected to Western blotting.

### 4.20. SILAC Data Analysis

We downloaded the list of proteins associated with the nucleolus (supported by experimental evidence) from the human proteome atlas (https://www.proteinatlas.org/humanproteome/cell/nucleoli, accessed on 15 January 2022) and intersected with the interactome of HMGXB4 in the presence and absence of SB. A fold-change cutoff of 0.5 with *p* < 0.05 was set to identify proteins interacting with HMGXB4. Enrichment analysis was conducted using GeneGo curated ontologies and Gene Ontology to provide a quantitative analysis of the data’s most relevant biological or molecular functions.

### 4.21. Single-Cell RNA-Seq Analysis

Single-cell (sc)RNA-seq datasets were downloaded in a raw format from five independent studies: human embryogenesis (GSE36552), mouse embryogenesis (GSE45719), differentiation of pluripotent stem cells to human primordial germ-like cells (hPGC-like) (GSE102943), hPGCs in a given space and time (GSE86146), and development of human germline cells in a gonadal niche (GSE86146). The datasets comprised ~3500 single-cell transcriptomes. The transcription of genes in every cell was calculated at TPM or FPKM expression levels. Samples were included in the analysis only if they had gene expression data of at least 5000 genes with expression levels exceeding the defined threshold (Log2 TPM > 1). We considered only those genes for the analysis that were expressed in at least 1% of the total samples. We used Seurat 1.2.1 from R to normalize the datasets at logarithmic scale using “scale.factor = 10,000”. After normalization, we calculated scaled expression (z-scores for each gene) for downstream dimension reduction. The cells were separated by subjecting the MVGs ({Log(Variance) and Log2(Average Expression)} > 2) to the dimension reduction methods of principal component analysis (PCA), which were further subjected to TSNE analysis. Consequently, HMGXB4 expression from normalized data in each cluster was calculated and visualized using the same tool.

### 4.22. Data Mining of ChIP-Seq Datasets

ChIP-seq datasets for histone modifications were obtained from the ENCODE project. ERK2/MAPK1/ELK1 ChIP-seq data were obtained from [44]. H3K27ac, MED1, POU5F1, POLII and CTCF ChIP-seq datasets were obtained from (GSE69646) [57]. Hi-C data was obtained from [47] (GSE116862) in “.hic” format. These contact matrix files were further normalized and visualized as a heatmap using the “Juicer” tool. ChIP-seq datasets in raw fastq format were aligned against the hg19 reference genome by Bowtie (version 2.2.2) under the parameters “local-sensitive”. All unmapped reads, non-uniquely mapped reads and PCR duplicates were removed from the analysis. Aligned reads were converted into bedGraph format using genomeCoverageBed from BedTools to visualize utilizing IGV over RefSeq genes (hg19).

### 4.23. Data Mining of ChIP-Exo Peaks of KRAB-ZNF Proteins

ChIP-exo peaks of 230 KRAB-ZNF protein were obtained from GSE78099. Signals from ChIP-exo data were obtained from MACS2 (as in [51]) after running over with parameter *-g hs -q 0.01 -B*. To visualize the overlapping peaks at the HMGXB4 locus, the obtained signals were merged beneath the Integrative genome visualizer (IGV) tracks. The density of MACS2 signals was plotted around the TSS of HMGXB4 to show the significant occupancy.

### 4.24. Statistics

Statistical analyses (Student’s *t*-test) were performed using the Prism 5 software https://www.graphstats.net/ (GraphPad) for molecular biology experiments. R was used for the statistical analysis of high throughput data. The significance level was calculated using the Wilcoxon test and corrected by multiple testing.

## Figures and Tables

**Figure 2 ijms-24-07283-f002:**
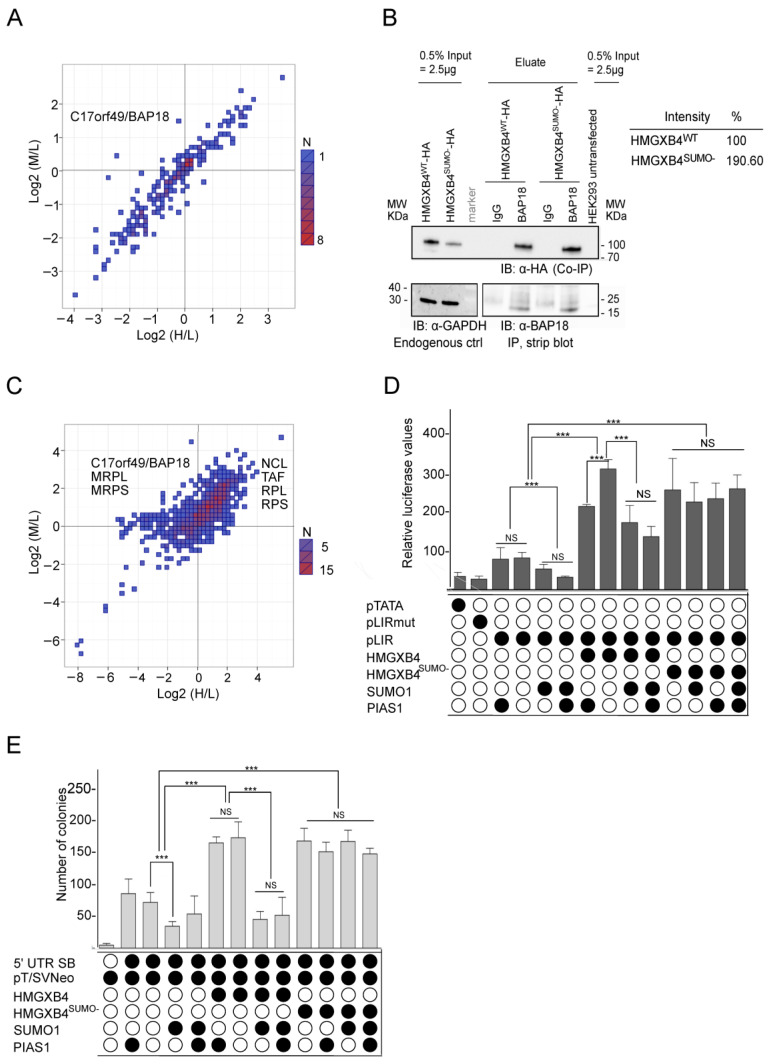
SUMOylation Interferes with the Transcriptional Activator Function of HMGXB4. (**A**) SILAC on HMGXB4 pull-down shows the interactome of SUMOylated and non-SUMOylated versions of HMGXB4 from HEK293 cells. Scatter diagram showing the comparison between the two conditions. The *X*-axis represents the Log2-fold change of SUMOylated HMGXB4 vs. controls (H/L), whereas the *Y*-axis represents the Log2-fold change of non-SUMOylated HMGXB4 vs. controls (M/L). See Appendix A for the experimental design and the description of Heavy, Medium and Light conditions. Note that C17orf4/BAP18 is detected in the interactome of the non-SUMOylated HMGXB4. (**B**) Representative co-immunoprecipitation to validate HMGXB4 and C17orf49/BAP18 interaction in HEK293 cells. (Right panel) Quantification of the intensity signals. Note that while both HMGXB4WT and HMGXB4^SUMO-^ interact with BAP18, the non-SUMOylated HMGXB4^SUMO-^ captures more BAP18 protein molecules. (**C**) The SILAC interactome of HMGXB4 reveals that the interactions are more intense in the presence of the SB transposase. Scatter diagram showing the comparison between experimental conditions used in SILAC. The *X*-axis represents the Log2-fold change of SUMOylated HMGXB4^WT^ vs. controls (H/L), whereas the *Y*-axis represents the Log2-fold change of non-SUMOylated HMGXB4^SUMO-^ vs. controls in the presence of the SB transposase (M/L). See Appendix A for the experimental design and the description of Heavy (H), Medium (M) and Light (L) conditions. The most significant differentially enriched proteins in the interactomes of non-SUMOylated HMGXB4^SUMO-^ and SUMOylated HMGXB4^WT^ and the presence of the SB transposase are shown. (**D**) SUMOylation interferes with the transcriptional activator function of HMGXB4. The effect of SUMO1 and PIAS1 on SB transposase expression is shown using a Dual-Luciferase Reporter assay. Luciferase reporter assays were performed using HeLa cell lysates, where reporter constructs were transiently co-transfected with wild-type or mutant HMGXB4 and PIAS1 in various combinations. The luciferase reporter plasmids were either controlled by TATA-box (pTATA) or SB 5′UTR promoter (pLIR LUC). pΔLIR LUC lacks the HMGXB4 response motif, and a minimal promoter (TATA-box) was used as a control. Expression constructs for PIAS1, SUMO1, wild-type HMGXB4 and SUMOylation mutant of HMGXB4^SUMO-^ were used. Standard errors of the mean are from three independent transfections (*** *p* < 0.01). (**E**) Barplots show that SUMOylation mitigates the transposition activator function of HMGXB4 and the effect of SUMO1 and PIAS1 expression on SB transposition. The transposase (driven by its promoter, 5′UTR SB) and the marker construct pT2/SVNeo are co-transfected with expression constructs of SUMO1 and PIAS1, wild-type HMGXB4 and SUMOylation mutant of HMGXB4^SUMO1-^ in various combinations into HeLa cells and subjected to a colony-forming transposition assay. As an additional control, we used a CMV-driven transposase expression construct, not regulated by HMGXB4. Standard errors of the mean are from three independent transfections (*** *p* < 0.01).

**Figure 3 ijms-24-07283-f003:**
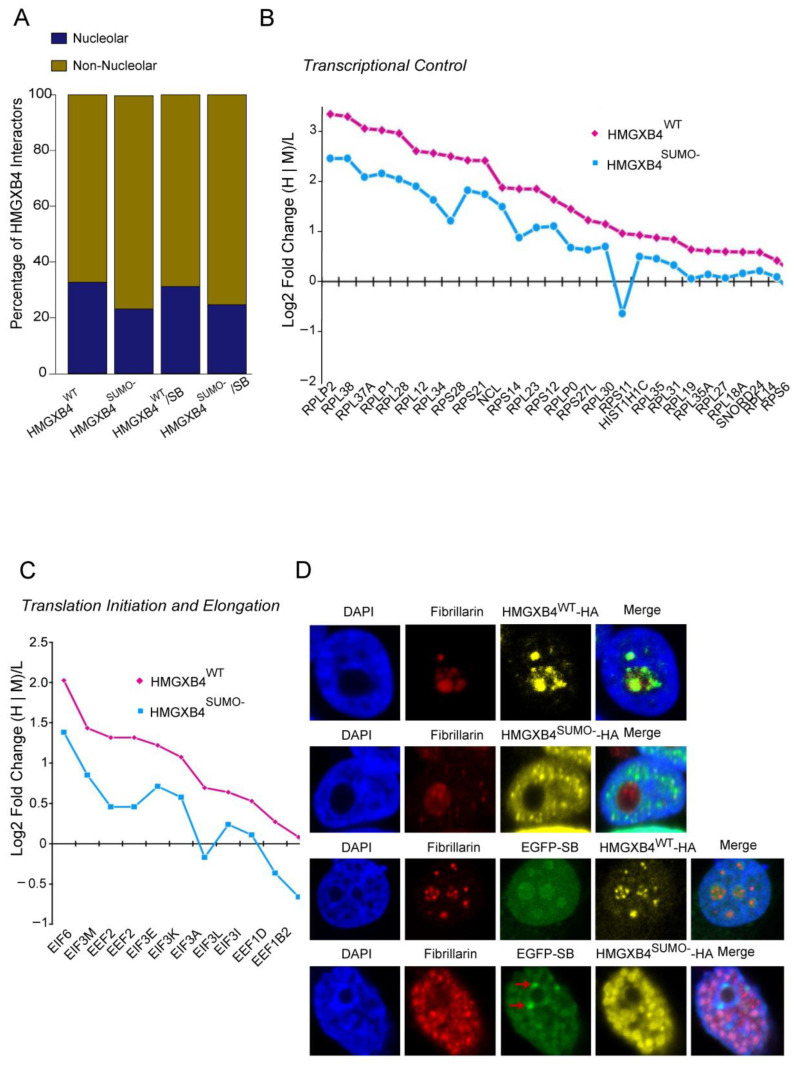
SUMOylated HMGXB4 is Compartmentalized to the Nucleolus. (**A**) SUMOylation affects the affinity of HMGXB4 to its nucleolar interacting partners. The stacked bar plot visualizes the differential affinity of HMGXB4 and HMGXB4^SUMO-^ with nucleolus-associated proteins in the presence and absence of the SB transposase. (**B**) Line plots display the effect of SUMOylation on the affinity of HMGXB4 to a selected set of protein-interacting partners involved in *Transcriptional control.* Note that the depicted proteins are reported to be localized in the nucleolus (by experimental evidence). (**C**) Line plots display the effect of SUMOylation on the affinity of HMGXB4 to a selected set of protein-interacting partners involved in *Translation initiation and elongation.* Note that the depicted proteins are reported to be localized in the nucleolus (by experimental evidence). (**D**) Visualization of subnuclear localization of transiently expressed HMGXB4^WT^ or HMGXB4^SUMO-^ (HA-tagged versions), using immuno-fluorescent confocal microscopy in HeLa cells. Upper two rows; HMGXB4^WT^-HA (yellow); fibrillarin (red); DAPI (blue); merge; Lower two rows; as on the upper panels, but in the presence of transiently co-expressed SB transposase (EGFP-SB, green). Note that the SB transposase co-localizes with both HMGXB4^WT^ and HMGXB4^SUMO-^, except in the perinuclear nuage (red arrows). The co-expression of HMGXB4^SUMO-^ and SB transposase disrupts the integrity of the nucleolus, which mobilizes the fibrillarin marker all over the cytoplasm. (Scale bars 40 µM).

**Figure 4 ijms-24-07283-f004:**
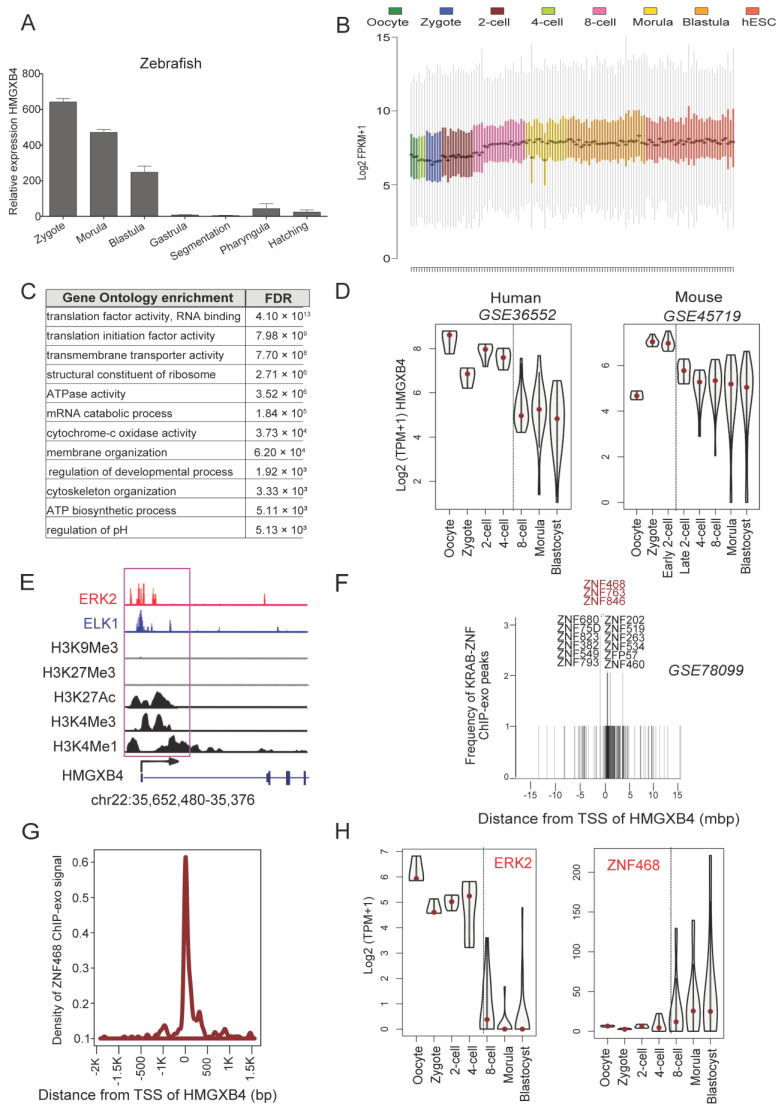
Transcription regulation of HMGXB4 in early development. (**A**) Transcription of HMGXB4 in developing zebrafish embryos (~100) collected at various stages of development, monitored by qRT-PCR (*n* = 2, normalized to GAPDH). (**B**) Boxplots showing the expression distribution of 413 genes that are highly expressed (Log2 FPKM > 2) in every single cell (>99%) of human preimplantation embryos. Besides known housekeeping genes (e.g., GAPDH and actin), HMGXB4 is expressed in every cell. (**C**) Top Gene Ontology (GO) of the 413 ubiquitously expressed genes. GO identification numbers from top to bottom. GO:0008135, GO:0003743, GO: 0022857, GO:0003735, GO:0016887, GO:0000184, GO:0004129, GO:0061024, GO:0050793, GO:0007010, GO:0006754, GO:0006885. (**D**) HMGXB4 is expressed before embryonic genome activation (EGA) at the highest level in both humans and mice. Violin plots display the Log2 normalized expression of HMGXB4 in the preimplantation embryos of humans (left) and mice (right) from the analysis of single-cell scRNA-seq datasets. TPM, transcripts per million). (**E**) HMGXB4 is controlled by a dual occupancy of ERK2/ELK1 transcription factors in pluripotent stem cells. Integrative Genomic Visualization (IGV) of various ChIP-seq normalized signals over the HMGXB4 locus in H1-ESCs. The ERK2/ELK1 DNA binding and chromatin mark signals from Chip-seq analysis over the HMGXB4 transcription start site region (black arrow) are highlighted (purple box). Histone modification datasets are from the ENCODE project, whereas ERK2 and ELK1 ChiP-seq datasets are from [44]. (**F**) Histogram illustrates the distance of 230 KRAB-ZNF ChIP-exo peaks from the Transcription Start Site (TSS) of HMGXB4. ZNF468, ZNF763 and ZNF846 (red) peaks (MACS2, adjusted *p* < 0.01, BH corrections) intersected on HMGXB4 TSS. Note that only the expression of ZNF468 showed a robust anti-correlation with HMGXB4 expression, thus predicting ZNF468 as a repressor of HMGXB4. (**G**) KRAB-ZNF468 binding interferes with the transcription of HMGXB4. The line plot shows the density of the KRAB-ZNF468 raw ChIP-exo signal over the TSS of HMGXB4. (**H**) Similar (e.g., activation) and antagonistic (e.g., repression) expression of ERK2/MAPK1 and ZNF468 with HMGXB4. Violin plots display the Log2 normalized expression of ERK2/MAPK1 and ZNF468 in the preimplantation embryos of humans from the analysis of single-cell RNA-seq datasets. Note that the expression of ZNF468 showed anti-correlation with HMGXB4 expression. EGA, embryonic gene activation.

**Figure 6 ijms-24-07283-f006:**
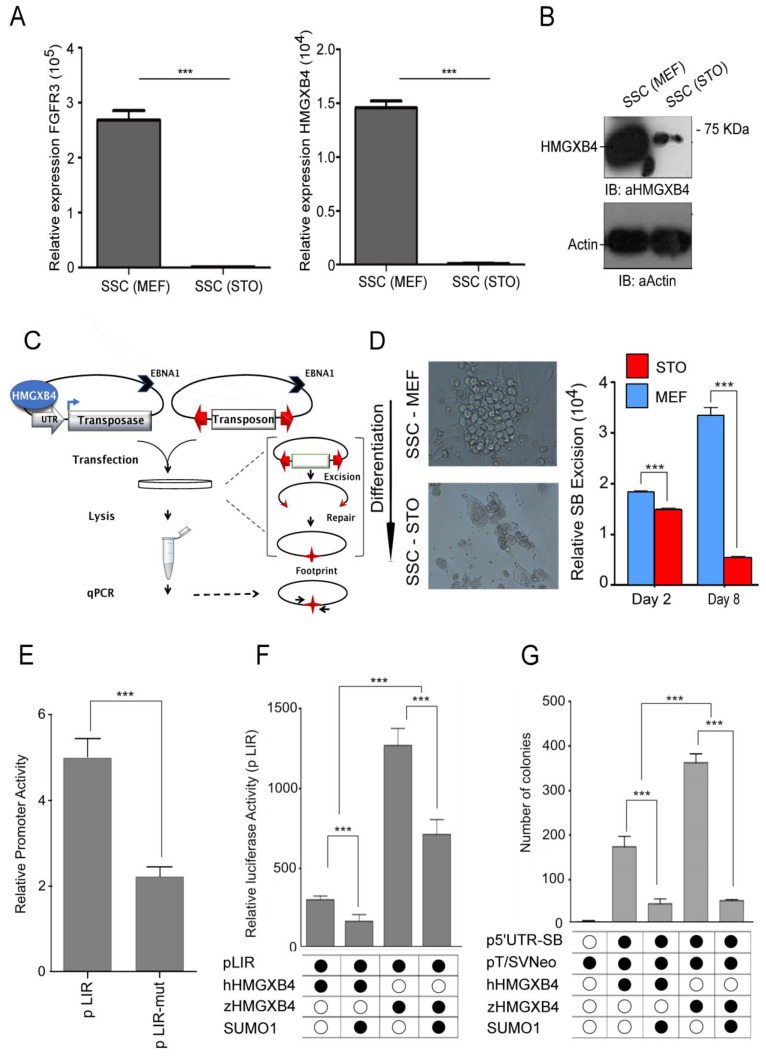
HMGXB4 Targets *Sleeping Beauty* Transposition to the Germline. (**A**) HMGXB4 transcription in spermatogonial stem cells (SSCs) declines upon differentiation (qRT-PCR). Relative expression FGFR3 (left panel) and HMGXB4 (right panel) in rat SSCs cultured on either MEF or STO feeder cells (*** *p* < 0.01). (**B**) HMGXB4 protein level is reduced in rat SSCs when cultured on STO feeder cells. Whole-cell lysates of SSC cultured with MEF and STO cells were subjected to immunoblotting using an anti-HMGXB4 antibody. (Normalized to GAPDH, *** *p* < 0.01). (**C**) Schematic of the quantitative transposon excision assay. Both the plasmid-based transposon and the transposase constructs have an EBNA1 gene providing replication in eukaryotic cells (inhibits degradation of the construct). The 5′UTR promoter of the SB transposon drives transposase expression and HMGXB4 binds the 5′ regulatory region of SB, resulting in enhanced transposase expression [10]. The transposon is flanked by terminal IRs (red arrows), carrying recognition sequences for the transposase. Following transfection into SSCs, the transposase excises the transposon, leaving a footprint (red star) behind. The footprint is quantifiable using qPCR. (**D**) (Left panel) Microscopic capture of undifferentiated and differentiated spermatogonial stem cells (SSCs) from the testes of rats. (20 × magnification). SSC grown on mouse embryonic feeders (MEF) remains undifferentiated, whereas it differentiates by replacing MEF with STO [56] (8 days). (Right panel) Excision of the SB transposon declines upon SSC differentiation. The quantitative transposon excision assay detects transposon excision from transiently transfected cells on days two and eight. Continued culturing on MEF (blue) replacing MEF to STO (red). Transposon excision was performed in three biological replicates. Average Ct values were calculated from quadruplicates of each sample (****p* < 0.01). (**E**) The activity of the 5′UTR regulatory region of the SB transposon with/without the HMGXB4 responding region in zebrafish embryos. One-cell stage zebrafish embryos (~100) were microinjected with a luciferase reporter construct driven by the 5′UTR promoter of SB (located in the left IR, pLIR). A deleted version of the inverted repeat (pΔLIR), not responding to HMGXB4 [10], was used as a control. Thirty-six hours after microinjection, a Dual-Luciferase reporter assay was performed (*** *p* < 0.01). (**F**) Comparison of the transcription modulating effect of HMGXB4 of either (zebra)fish or human origin. HeLa cells were transiently co-transfected with a luciferase reporter construct under the control of the SB 5′UTR promoter (located in the left IR, pLIR) and expression plasmids for zHMGXB4 or hHMGXB4 and SUMO1. Cells were harvested and analyzed for luciferase activity 48 h post transfection c (*** *p* < 0.01). (**G**) Comparison of the effect of HMGXB4 expression of either (zebra)fish or human origin on SB transposition. HeLa cells were co-transfected with a neo reporter construct, pT/SVNeo, an SB transposase expression construct under the control of SB 5′UTR promoter (located in the left IR, pLIR) and zHMGXB4 or hHMGXB4 in the presence and absence of SUMO1 into HeLa cells, and were subjected to a colony formation (transposition) assay [11] *** (*p* < 0.01).

**Table 1 ijms-24-07283-t001:** PCR primers used in the quantitative transposon excision assay.

forAS2	5′-TATGACCATGATTACGCCAAGCT-3′
revAS1	5′-CGACGGCCAGTGAATTCG-3′
probeAS1	5′-TCTAGAGGATCCCCTACWGTAGGTACCG-3′ W = A/T
ampB for	5′- GTGTCGCCCTTATTCCCTTTT-3′
ampB rev	5′-TGCGGCATTTTGCCTTCC-3′
ampB probe	5′-GCGTTTCTGGGTGAGCAAA-3′

Note: For the ambiguous base in the footprint, a probe was ordered with a degenerate nucleotide in that position to pick up both variations in our PCR.

## Data Availability

The raw SILAC data of HMGXB4 interactome and the Y2H data are available in the Appendix A.

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
