# Peer review of "HMGXB4 Targets Sleeping Beauty Transposition to Germinal Stem Cells"

_ijms, 2023, doi:10.3390/ijms24087283_

Round 1

Reviewer 1 Report

 In this study, Devaraj and colleagues suggest that subcellular localization of HMGXB4 is mediated by post-translational SUMOylation and predominantly expressed in the early-embryogenesis. This expression is suggested to be associated with the activation of Sleeping Beauty Transposition by targeting its 5’UTR promoter region.

Initial Y2H and coIP analyses suggest that SUMO1 is an interaction partner and responsible for mono/poly SUMOylation of HMGXB4. The high-throughput SILAC mass spectrometry analysis and coIP suggest having a potential interaction with BAP18 chromatin reading complex member. The SUMOylation of HMGXB4 was suggested to interfere with the transcriptional activator function of HMGXB4, shown by an in vitro reporter assay. Moreover, immunostaining analyses suggested a potential subcellular switch regulatory effect of SUMOylation which compartmentalized the HMGXB4 into the nucleolus. Further, public ChIP-Seq data analysis suggested potential association with stemness whose promoter was active in early-embryogenesis stages. Ultimately, by an in vitro transposition reporter assay, it is suggested that HMGXB4 binds to the promoter region of Sleeping Beauty Transposone and drives its expression in germline cells.

It is an extensively investigated study, having a core original question, and at high potential interest to the readers. However, it is experiencing some shortcomings and also lacking of some essential controls.

For that, the authors are kindly asked to consider the following major and minor points to improve the manuscript.

Major points:

1- In SUMO1 condition (Fig.1B), there is a distinct band and a smear of sumoylated HMGXB4. It seems like mono- and poly-sumoylation events are taking place when SUMO1 is in action. However, the smear disappears when SUMO2 or SUMO3 is overexpressed. Could authors please explain what kind of sumoylation event is interfering with the HMGXB4 activation?

2- The authors suggest that sumoylation is important in the mediation of the interaction between HMGXB4 and BAP18 according to the current high throughput MS data and previous literature. However, its essential to add an extra condition (sumoylation deficient HMGXB4) into co-IP experiment (Fig.2B) to verify this finding and conclusion.

3- The statement (Lines 281-282) about the colocalization of HMGXB4SUMO- with SB in nucleoplasm is not strongly supported by the colocalization image presented in Fig.3C. It seems that both HMGXB4SUMO- and EGFP-SB generate nuclear puncta but not showing clear colocalization compared to wt counterpart of HMGXB4, showing clear colocalization with SB in the nucleolus. The authors are suggested to either provide a better representative image or correct the statement.

4- As it is shown in Fig.4A, the expression of HMGXB4 was drastically declined in the late phases of embryogenesis. To be able to make a correct comparison and estimation about the potential effects of ERK2, ELK1, as well as KRAB-ZNF/TRIM28 system in early embryogenesis using publicly available ChIP-Seq and ChIP-exo Seq data (Fig.4E-G), authors should also have some control ChIP-Seq and ChIP-exo Seq data; something like non-stem cells and non-embryonic stem cells.

5- How is the subcellular localization of HMGXB4wt and HMGXB4SUMO- in stem cells compared to non-stem cells?

6- What is the functional relevance of HMGXB4 in the maintenance of stemness? What could be the consequence of the suppression of HMGXB4 in stem cells?

7- Is HMGXB4 indispensable for the activation of SB in germlines? In the absence of HMGXB4, would authors expect any other HMG family member to take over the role? Is there any other HMG family member, or any transcription factor, which is able to recognize the same DNA sequence at the promoter of SB?

Minor points:

1- The text on Fig.2B requires correction: Which of the proteins immunoprecipitated/immunoblotted?

2- All the figures are in low resolution. It seems like a document conversion issue. The figures’ quality should be optimal in the revised version of the manuscript.

3- The authors are highly encouraged to upload the raw Y2H and raw SILAC mass spec data to a public repository.

4- Unfortunately, there is no supplementary method docx file uploaded in the SuSy system.

Author Response

In this study, Devaraj and colleagues suggest that subcellular localization of HMGXB4 is mediated by post-translational SUMOylation and predominantly expressed in the early-embryogenesis. This expression is suggested to be associated with the activation of Sleeping Beauty Transposition by targeting its 5’UTR promoter region.

Initial Y2H and coIP analyses suggest that SUMO1 is an interaction partner and responsible for mono/poly SUMOylation of HMGXB4. The high-throughput SILAC mass spectrometry analysis and coIP suggest having a potential interaction with BAP18 chromatin reading complex member. The SUMOylation of HMGXB4 was suggested to interfere with the transcriptional activator function of HMGXB4, shown by an in vitro reporter assay. Moreover, immunostaining analyses suggested a potential subcellular switch regulatory effect of SUMOylation which compartmentalized the HMGXB4 into the nucleolus. Further, public ChIP-Seq data analysis suggested potential association with stemness whose promoter was active in early-embryogenesis stages. Ultimately, by an in vitro transposition reporter assay, it is suggested that HMGXB4 binds to the promoter region of Sleeping Beauty Transposone and drives its expression in germline cells.

It is an extensively investigated study, having a core original question, and at high potential interest to the readers. However, it is experiencing some shortcomings and also lacking of some essential controls.

For that, the authors are kindly asked to consider the following major and minor points to improve the manuscript.

Major points:

1- In SUMO1 condition (Fig.1B), there is a distinct band and a smear of sumoylated HMGXB4. It seems like mono- and poly-sumoylation events are taking place when SUMO1 is in action. However, the smear disappears when SUMO2 or SUMO3 is overexpressed. Could authors please explain what kind of sumoylation event is interfering with the HMGXB4 activation?

Response: Our data suggest that the HMGXB4 protein can be modified by all three SUMO variants (SUMO1-3). However, when SUMO1 is in action, we observe mono- and poly-SUMOylation events, whereas in the presence of SUMO2 or SUMO3, mono-SUMOylation products are likely to be formed. With our in vitro SUMOylation assays, we also observed that both SENP1 and SENP2 reduced the conjugation of SUMO1, while SENP3 deconjugated SUMO2 (Figure 1D-E). Based on these observations, we hypothesise that post-translational modifications by SUMO1, SUMO2 and SUMO3 might regulate distinct biological processes. In the current study, we only followed the SUMO1 modification.

2- The authors suggest that sumoylation is important in the mediation of the interaction between HMGXB4 and BAP18 according to the current high throughput MS data and previous literature. However, its essential to add an extra condition (sumoylation deficient HMGXB4) into co-IP experiment (Fig.2B) to verify this finding and conclusion.

Response: Thank you for this comment. Following the reviewer’s suggestion, we repeated the co-IP experiment, now including the SUMOylation deficient HMGXB4 (revised Fig. 2). We were able to demonstrate an interaction with both the wt and mutant versions, but the Co-IP band was stronger with the SUMOylation deficient HMGXB4 mutant. This is consistent with the MS data. BAP18 has already been shown to interact with wild-type HMGXB4 in the literature, suggesting that it is not an absolute difference. Overall, the results suggest that the interaction between HMGXB4 and BAP18 is influenced by the stoichiometry of HMGXB4 modulated by SUMOylation, which is altered by subcellular trafficking. Please also note that the presence of the endogenous wild-type HMGXB4 can also be detected by co-IP.

3- The statement (Lines 281-282) about the colocalization of HMGXB4SUMO- with SB in nucleoplasm is not strongly supported by the colocalization image presented in Fig.3C. It seems that both HMGXB4SUMO- and EGFP-SB generate nuclear puncta but not showing clear colocalization compared to wt counterpart of HMGXB4, showing clear colocalization with SB in the nucleolus. The authors are suggested to either provide a better representative image or correct the statement.

Response: Thank you for pointing this out. Indeed, the images do not show clear co-localisation of HMGXB4SUMO- and EGFP-SB, and the staining signals do not overlap. The more precise statement is that SB transposase in the presence of HMGXB4WT is scattered in the nucleus, with some enrichment in the nucleolar compartment, whereas in the presence of HMGXB4SUMO- SB transposase is enriched in the nuclear nuage and undetectable in the nucleolus.

4- As it is shown in Fig.4A, the expression of HMGXB4 was drastically declined in the late phases of embryogenesis. To be able to make a correct comparison and estimation about the potential effects of ERK2, ELK1, as well as KRAB-ZNF/TRIM28 system in early embryogenesis using publicly available ChIP-Seq and ChIP-exo Seq data (Fig.4E-G), authors should also have some control ChIP-Seq and ChIP-exo Seq data; something like non-stem cells and non-embryonic stem cells.

Response: We thank the reviewer for pointing it out. The concern is specificity of DNA binding of the listed factors. We have calculated the ChIP-seq/exo signals relative to their respective controls, which is the total input and pulled-down DNA by IgG antibodies. The pipeline employed here detects the significant peaks (enrichment of sequencing reads over a given genomic window) only if the False Discovery Rate (FDR) is below 1% from the genome-wide comparisons against the controls. For the significance, we have calculated and reported the P-values. The analysis pipeline for these high-throughput datasets is the same as reported in our previous research article (https://elifesciences.org/articles/76257#sa2).

Here, we have discussed only those factors that are significantly enriched around the promoter of HMGXB4.

We have provided the brief details in method section line number 1501-1506.

We amend the following sentences in the manuscript for clarification.

  • Line number 492: This approach identified repressive KRAB-ZNF proteins (e.g. ZNF468, ZNF763 and ZNF846), harbouring significant peaks (adjusted p-value < 1e-7, compared with total input control) at the transcription start site (TSS) of HMGXB4.
  • Line number 506: This boundary is marked by CTCF (Figure 5A) and co-occupied by ChIP-seq peaks for H3K27ac, MED1 (Mediator 1), POU5F1/OCT4 and POLII (adjusted-p-value < 0.01, BH corrections) over the TSS of HMGXB4, connecting gene expression and chromatin architecture.
  • Line number 509: Adding additional layers of CUT&RUN data analysis for H3K4me3 and H3K27me3 uncovers that the TSS of HMGXB4 has an enrichment for H3K4Me3 (adjusted-p-value < 0.01, BH corrections) but not for H3K27Me3 in human germinal vesicle (GV) stage oocytes, 4-cell, 8-cell and ICM (inner cell mass).
  • Line number 923: Histogram illustrates the distance of 230 KRAB-ZNF ChIP-exo peaks from the Transcription Start Site (TSS) of HMGXB4. ZNF468, ZNF763 and ZNF846 (red) peaks (MACS2, adjusted-p-value < 0.01, BH corrections) intersected on HMGXB4 TSS. Note that only the expression of ZNF468 showed a robust anti-correlation with HMGXB4 expression, thus predicting ZNF468 as a repressor of HMGXB4.

5- How is the subcellular localization of HMGXB4wt and HMGXB4SUMO- in stem cells compared to non-stem cells?

Response: Among differentiated cells, HMGXB4 is most strongly expressed in germ cells of spermatogonia, followed by germ cells of spermatocytes (~80 nTPM) (Human Protein Atlas). Our own analysis of the expression data shows that HMGXB4 is primarily expressed in stem cells rather than differentiated cells, and that its expression is increased in several cancer cell lines (see figure).

6- What is the functional relevance of HMGXB4 in the maintenance of stemness? What could be the consequence of the suppression of HMGXB4 in stem cells?

Response: Our data suggest that although HMGXB4 is not part of the NURF core complex, when expressed it can modulate its function. Previous studies have identified HMGXB4 as a factor that inhibits WNT signalling [17][18]. Here we show that HMGXB4 functions as a transcription factor/co-activator of transposase SB. The exact role of HMGXB4 in stem cells was not investigated in the current study, but in a follow-up study. Our unpublished data suggest that suppression of HMGXB4 in embryonic stem cells activates the transcriptional program for mesendodermal lineages.

7- Is HMGXB4 indispensable for the activation of SB in germlines? In the absence of HMGXB4, would authors expect any other HMG family member to take over the role? Is there any other HMG family member, or any transcription factor, which is able to recognize the same DNA sequence at the promoter of SB?

Response: This is a very interesting point worthy of further investigation. Previously, we identified another member of the HMG family (HMGB1) as an interaction partner of the SB transposase doi: 10.1093/nar/gkg341. However, HMGB1 does not specifically recognise DNA and plays a different role in SB transposition (complex assembly). In addition, we have unpublished data that SRY (SRY sex determining region Y), which is a member of the High Mobility Group (HMG) box family of DNA-binding proteins similar to HMGXB4, also binds to the promoter of SB transposase, although it recognises a different sequence. HMGXB4 is therefore not the only factor that could transcriptionally activate SB transposition. SRY is the testis-determining factor (TDF) that initiates male sex determination. Whether HMGXB4 and SRY play an overlapping/complementary role in SB transposition needs to be investigated in future studies. Currently, we have no strong evidence to support either scenario.

Minor points:

1- The text on Fig.2B requires correction: Which of the proteins immunoprecipitated/immunoblotted?

Response: A new experiment is provided (including the HMGXB4SUMO-) with corrected information.

2- All the figures are in low resolution. It seems like a document conversion issue. The figures’ quality should be optimal in the revised version of the manuscript.

Response: Indeed, it was a conversion issue. High quality figures would be provided for the Journal.

3- The authors are highly encouraged to upload the raw Y2H and raw SILAC mass spec data to a public repository. 4- Unfortunately, there is no supplementary method docx file uploaded in the SuSy system.

Response: The submission process of the raw SILAC data to a public depository is in process (NCBI needs four more days to complete, and the reference number will be communicated to the Journal). The Y2H data are ready to be uploaded and available as a supplementary method docx file. 

Reviewer 2 Report

Here, the manuscript titled asHMGXB4 Targets Sleeping Beauty Transposition to Germinal Stem Cellsby Anantharam Devaraj et al., characterized the model of HMGXB4 activate the SB transposition in germinal stem cells, and the expression in zebrafish embryos, mice and human pre-implantation embryos, defined the regulation pathway of HMGXB4 involved, and presented data suggested that HMGXB4 Links Pluripotent and Germinal Stem Cells. Overall, the paper is well written, and the methodology for the most part seems sound.  I have only a few minor concerns and suggestions I would like to see addressed prior to publication.

1.      How about the application potential of the activation of SB transposition by HMGXB4 in vertebrate transgenesis, is it possible to apply this protein as a co-factor with transposase to increase the transgenesis efficiency in zebrafish and mice. Considering SB is popularly applied genetic tool in transgenesis, and discussion will be highly appreciated.

2.      How many repeats were calculated for transposition excision by qPCR in Fig.6D, please add the errors bar in figure 6d, and add the repeat number for qPCR in each test in methods.

3.      The statement “The aberrant activation of this tightly controlled developmental factor might result in undesirable target gene expression, suggesting that HMGXB4 could be an important target in cancer and inflammation research.” in abstract is largely deduced, and is somehow far away from the topic of MS, please reword, may delete it.

4.      An overall conclusion of the whole MS in the end may help for understanding.

Author Response

Here, the manuscript titled as” HMGXB4 Targets Sleeping Beauty Transposition to Germinal Stem Cells” by Anantharam Devaraj et al., characterized the model of HMGXB4 activate the SB transposition in germinal stem cells, and the expression in zebrafish embryos, mice and human pre-implantation embryos, defined the regulation pathway of HMGXB4 involved, and presented data suggested that HMGXB4 Links Pluripotent and Germinal Stem Cells. Overall, the paper is well written, and the methodology for the most part seems sound.  I have only a few minor concerns and suggestions I would like to see addressed prior to publication.

  1. How about the application potential of the activation of SB transposition by HMGXB4 in vertebrate transgenesis, is it possible to apply this protein as a co-factor with transposase to increase the transgenesis efficiency in zebrafish and mice. Considering SB is popularly applied genetic tool in transgenesis, and discussion will be highly appreciated.

Response: Thank you very much for the question. Indeed, the SB transposon system has been successfully established as a transgenic tool (doi: 10.1038/nprot.2014.010; doi: 10.1038/nprot.2014.009; doi: 10.1038/nprot.2014.008; doi: 10.1096/fj.12-205526; doi: 10.1002/dvdy.23891). For such an application, we recommend the injection of the transposase encoding mRNA together with the transgene (flanked by the inverted repeats of the SB containing the binding sites for the SB transposase) into the embryo at the one-cell stage. A similar protocol has been successfully used in mice, rats, pigs and mammals. The HMGXB4-mediated regulatory region is located in the transposase promoter. Since the transposase mRNA is used in the transgenic protocols, the HMGXB4-mediated regulatory region is irrelevant in the transgenic protocols.

  1. How many repeats were calculated for transposition excision by qPCR in Fig.6D, please add the errors bar in figure 6d, and add the repeat number for qPCR in each test in methods.

Response: Transposon excision was performed in three biological replicates. Average Ct values were calculated from quadruplicates of each sample. Error bars and significance are added.

  1. The statement “The aberrant activation of this tightly controlled developmental factor might result in undesirable target gene expression, suggesting that HMGXB4 could be an important target in cancer and inflammation research.” in abstract is largely deduced, and is somehow far away from the topic of MS, please reword, may delete it.

Response: The sentence is removed from the abstract.

  1. An overall conclusion of the whole MS in the end may help for understanding.

Response: We have added on the line number 762: Overall, our study highlights HMGXB4 as an evolutionarily conserved host-encoded factor that assists Tc1/Mariner transposons target the germline, which was necessary for their fixation and may explain their abundance in vertebrate genomes.

Round 2

Reviewer 1 Report

The authors revised and significantly improved the manuscript by addressing the questions and by taking into consideration the suggestions. Thus, the current version of the manuscript is suggested for publication in IJMS.